# Causes, characteristics, and patterns of prolonged unplanned school closures prior to the COVID-19 pandemic—United States, 2011–2019

**Ferdous A. Jahan**[1,2], **Nicole Zviedrite**[1]*, **Hongjiang Gao**[1], **Faruque Ahmed**[1], **Amra Uzicanin**[1]

**1** Centers for Disease Control and Prevention, Atlanta, Georgia, United States of America, **2** Cherokee Nation Operational Solutions, LLC, Tulsa, Oklahoma, United States of America

☯ These authors contributed equally to this work.

* NZviedrite@cdc.gov

## Abstract

### Introduction

Outside of pandemics, there is little information about occurrence of prolonged unplanned K-12 school closures (PUSC). We describe here the reasons, characteristics, and patterns of PUSC in the United States during 8 consecutive inter-pandemic academic years, 2011–2019.

### Methods

From August 1, 2011 through June 30, 2019, daily systematic online searches were conducted to collect data on publicly announced unplanned school closures lasting ≥1 school days in the United States. Closures were categorized as prolonged when schools were closed for ≥5 unplanned days (approximating one full workweek), excluding weekends and scheduled days off per school calendars.

### Results

During the eight academic years, a total of 22,112 PUSCs were identified, affecting over 800,000 teachers and 13 million students that resulted in 91.5 million student-days lost. A median of 62.9% of students in PUSC-affected schools were eligible for subsidized school meals. Most affected schools were in cities (35%) and suburban areas (33%). Natural disasters (47%), adverse weather conditions (35%), and budget/teacher strikes (15%) were the most frequently cited reasons for PUSC; illness accounted for 1%, and building/facility issues, environmental issues and violence together accounted for the remaining 2%. The highest number of PUSCs occurred in Health and Human Services Regions 2, 3, 4, and 6 encompassing areas that are frequently in the path of hurricanes and tropical storms. The majority of PUSCs in these regions were attributed to a handful of hurricanes during the fall season, including hurricanes Sandy, Irma, Harvey, Florence, and Matthew.

**Data Availability Statement:** All relevant data are available from the Data.CDC.gov database (https://data.cdc.gov/Public-Health-Surveillance/

Prolonged-Unplanned-School-Closures-USA-2011-2019/5iuf-feyd).

**Funding:** This study was supported by the United States Centers for Disease Control and Prevention (http://www.cdc.gov/). The co-authors are or were employees (NZ HG FA AU) or contractors (FJ) of the US CDC at the time of the study. Ferdous Jahan (FJ) is employed by Cherokee Nation Operational Solutions, LLC. The funder (Cherokee Nation Operational Solutions) provides support in the form of salary for the author (FJ), but did not have any additional role in the study design, data collection and analysis, decision to publish, or preparation of the manuscript. The specific roles of all authors are articulated in the 'Author Contributions' section.

**Competing interests:** Ferdous Jahan is employed by Cherokee Nation Operational Solutions, LLC. This does not alter our adherence to PLOS ONE policies on sharing data and materials.

## Conclusions

PUSCs occur annually in the United States due to a variety of causes and are associated with a substantive loss of student-days for in-school learning. Both these prior experiences with PUSCs and those during the current COVID-19 pandemic illustrate a need for creating sustainable solutions for high-quality distance learning and innovative supplemental feeding programs nationwide, especially in disaster-prone areas.

## Introduction

During the school year, approximately 56.4 million students and 3.7 million teachers congregate every weekday across more than 100,000 Kindergarten-12th grade (K-12) schools across the US [1]. Beyond fostering intellectual as well as emotional and social growth, K-12 schools in the US also serve other important societal functions including the provision of childcare, school-based meals, special education, counseling, and other services [2]. When schools must close in an unplanned fashion, i.e., outside of the established academic calendar, interruption of the traditional education process occurs coupled with other unwanted consequences for students, families, and schools. Identifying the causes, patterns, and characteristics of prolonged unplanned school closures (PUSCs) can serve to inform preparedness in order to mitigate the economic and social consequences, which may disproportionately affect some students and their families.

In 2011, we initiated prospective daily data collection of publicly announced unplanned school closures lasting ≥1 days to better understand the community experience with the unplanned school closures occurring outside of influenza pandemics [3]. Using the same method, the data collection on all-cause closures was continued through June 30, 2019. Here we present an analysis of the causes, characteristics, and patterns of prolonged school closures occurring during the 8-year period from 2011–2019, which closely predates the late 2019 emergence of the severe acute respiratory syndrome coronavirus 2 (SARS-CoV-2) and subsequent coronavirus disease 2019 (COVID-19) pandemic.

## Materials and methods

### Data collection

Data were collected on unplanned school closures from August 1, 2011 to June 30, 2019, for K-12 schools and school districts in the United States (50 states and D.C.). Daily (Monday through Friday) systematic online searches of Google (Alerts, Search, and News) and Lexis-Nexis were conducted to capture publicly announced unplanned school or district closures lasting ≥1 school days using the data collection methodology detailed by Wong et al. [3]. Data were entered into a Microsoft Access database. For each incidence of school or district closure, the duration of closure was computed from the date of closure to the date of reopening, excluding any planned days off, such as weekends or school holidays. Schools or districts with more than one incidence of closure during the study period were counted separately for each closure. A closure duration of ≥5 days was categorized as PUSC, which represented at least 1 week of missed class time. The reason for PUSC was classified into the following categories: weather (e.g., ice or snowstorm, rainstorm, fog, wind, extreme temperature), natural disaster (e.g., hurricane, tornado, flood, wildfire, earthquake), building/facility issue, environmental problem, budget/teacher strike, violence, and illness.

Eight academic years of school district or school closure data (2011–2012 to 2018–2019) were linked with data for the corresponding year from the National Center for Education Statistics (NCES) [4, 5] using district and school NCES IDs. From the NCES data, information was extracted on the total number of public schools (for school district), total student enrollment, the total number of teachers, grade levels, school locations, locale, and subsidized school meals (public schools/school districts only). Twenty individual schools that did not match to schools in the NCES database were categorized as public or private after confirmation by online searches with their web addresses. Districts with zero schools; schools with zero students; vocational, special education, and alternative schools with missing students; online/virtual schools; adult education programs; summer schools; jails; and schools with pre-kindergarten or transitional kindergarten as the highest grade were excluded from the analysis. Schools were categorized based on the NCES-defined grade span, and were primarily organized into three levels, elementary (K– 5), middle (6–8), and high (9–12) schools. Schools having more than one level were classified as elementary-middle (K– 8), elementary-high (K– 12), or middle-high (6–12) schools. Schools where the location or grade levels were undefined, were classified as not specified. The 10 regions defined by the U.S. Department of Health and Human Services (HHS) were used to assess geographic variation.

### Analysis

Descriptive findings are presented using the following units of analysis: 1) district-wide or individual school event (e.g., closure of a school district with multiple schools was counted as one event; closure of an individual school where there was no district-wide closure was counted as one event); 2) school; 3) teacher; 4) student; and 5) student-day. Student-days lost was calculated by multiplying the number of students in a school with PUSC with the duration of PUSC.

Student-days lost per 1,000 students per year was computed by dividing the average student-days lost per year due to PUSCs with the average number of students enrolled per year in all K-12 schools during the eight-year study period and then multiplying by 1000. The cumulative incidence of PUSC per 100 schools over the eight-year study period was computed by dividing the number of schools with PUSCs during 2011–2019 by the number of all K-12 schools and then multiplying by 100. The number of all K-12 schools was obtained from the National Center for Education Statistics (NCES) by summing the number of schools reported for each academic year (2011–2012 through 2018–2019) and dividing by eight.

The association between selected characteristics and school closures of $\geq 6$ days among all schools with PUSCs was assessed using schools as the unit of analysis. The cutoff of 6 days was used as it represents the median duration of PUSC. Logistic regression (PROC LOGISTIC, SAS, Version 9.4, Cary, NC) was used for computing both unadjusted and adjusted odds ratios. The dependent variable in the regression models was coded as 0 (closure of 5 days) and 1 (closure of $\geq 6$ days). Interactions were assessed.

The project underwent ethical review at the Centers for Disease Control and Prevention's Human Research Protections Office and was determined not to involve human subjects; it was therefore not subject to institutional review board review requirements.

### Results

Among 120,579 district-wide or individual school closure events with closure duration of $\geq 1$ days during the period August 2011 to June 2019, there were 2,259 (1.9%) PUSC events (Table 1). The 2,259 PUSC events corresponded to 22,112 school closures; 21,585 (97.6%) school closures were due to 1,732 district-wide closure events, and the remaining 527 (2.4%)

**Table 1. Characteristics of prolonged unplanned school closure (PUSC) by academic year, United States, 2011–2019[a],[b].**

| | | PUSC by Academic Year | | | | | | | |
|---|---|---|---|---|---|---|---|---|---|
| | **Total** | **2011–12** | **2012–13** | **2013–14** | **2014–15** | **2015–16** | **2016–17** | **2017–18** | **2018–19** |
| **Unit of analysis: District-wide or individual school event** | | | | | | | | | |
| Number of ≥1 day closure events | 120,579 | 4,063 | 16,816 | 18,177 | 10,931 | 8,605 | 13,042 | 26,780 | 22,165 |
| Number of PUSC events (% of ≥1 day closure events) | 2,259 (1.9) | 99 (2.4) | 502 (3.0) | 157 (0.9) | 156 (1.4) | 125 (1.5) | 133 (1.0) | 639 (2.4) | 448 (2.0) |
| Type of PUSC event, n (column %) | | | | | | | | | |
| District-wide | 1,732 (76.7) | 85 (85.9) | 379 (75.5) | 129 (82.2) | 126 (80.8) | 109 (87.2) | 101 (75.9) | 467 (73.1) | 336 (75.0) |
| Individual school | 527 (23.3) | 14 (14.1) | 123 (24.5) | 28 (17.8) | 30 (19.2) | 16 (12.8) | 32 (24.1) | 172 (26.9) | 112 (25.0) |
| **Unit of analysis: School** | | | | | | | | | |
| Estimated number of PUSCs[c], n (row %) | 22,112 | 770 (3.5) | 4,513 (20.4) | 996 (4.5) | 1,498 (6.8) | 2,495 (11.3) | 1,382 (6.3) | 7,215 (32.6) | 3,243 (14.7) |
| Median duration of PUSC, days (5th, 95th percentile) | 6 (5–11) | 6 (5–8) | 5 (5–10) | 5 (5–8) | 6 (5–12) | 5 (5–7) | 7 (5–16) | 7 (5–11) | 5 (5–19) |
| Type of PUSC, n (column %) | | | | | | | | | |
| District-wide | 21,585 (97.6) | 756 (98.2) | 4,390 (97.3) | 968 (97.2) | 1,468 (98.0) | 2,479 (99.4) | 1,350 (97.7) | 7,043 (97.6) | 3,131 (96.6) |
| Individual school | 527 (2.4) | 14 (1.8) | 123 (2.7) | 28 (2.8) | 30 (2.0) | 16 (0.6) | 32 (2.3) | 172 (2.4) | 112 (3.5) |
| School type, n (column %) | | | | | | | | | |
| Public | 21,784 (98.5) | 765 (99.4) | 4,412 (97.8) | 988 (99.2) | 1,485 (99.1) | 2,486 (99.6) | 1,367 (98.9) | 7,084 (98.2) | 3,197 (98.6) |
| Private | 328 (1.5) | 5 (0.7) | 101 (2.2) | 8 (0.8) | 13 (0.9) | 9 (0.4) | 15 (1.1) | 131 (1.8) | 46 (1.5) |
| School grade level, n (column %) | | | | | | | | | |
| Elementary school | 9,203 (41.6) | 394 (51.2) | 1,677 (37.2) | 366 (36.8) | 660 (44.1) | 1,183 (47.4) | 627 (45.4) | 2,996 (41.5) | 1,300 (40.1) |
| Elementary to middle school | 4,405 (19.9) | 68 (8.8) | 1,125 (24.9) | 230 (23.1) | 317 (21.2) | 388 (15.6) | 189 (13.7) | 1,487 (20.6) | 601 (18.5) |
| Elementary to high school | 584 (2.6) | 46 (6.0) | 57 (1.3) | 20 (2.0) | 45 (3.0) | 57 (2.3) | 29 (2.1) | 196 (2.7) | 134 (4.1) |
| Middle school | 3,072 (13.9) | 115 (14.9) | 551 (12.2) | 146 (14.7) | 173 (11.6) | 384 (15.4) | 233 (16.9) | 1,015 (14.1) | 455 (14.0) |
| Middle to high school | 1,021 (4.6) | 19 (2.5) | 203 (4.5) | 56 (5.6) | 59 (3.9) | 84 (3.4) | 61 (4.4) | 392 (5.4) | 147 (4.5) |
| High school | 3,638 (16.4) | 124 (16.1) | 840 (18.6) | 168 (16.9) | 239 (16.0) | 373 (15.0) | 231 (16.7) | 1,081 (15.0) | 582 (18.0) |
| Not specified | 189 (0.9) | 4 (0.5) | 60 (1.3) | 10 (1.0) | 5 (0.3) | 26 (1.0) | 12 (0.9) | 48 (0.7) | 24 (0.7) |
| Cause of PUSC, n (column %) | | | | | | | | | |
| Weather | 7,770 (35.1) | 623 (80.9) | 434 (9.6) | 854 (85.7) | 1,418 (94.7) | 2,092 (83.9) | 351 (25.4) | 531 (7.4) | 1,467 (45.2) |
| Natural disaster | 10,496 (47.5) | 13 (1.7) | 3,410 (75.6) | 31 (3.1) | 12 (0.8) | 311 (12.5) | 981 (71.0) | 4,320 (59.9) | 1,418 (43.7) |
| Budget/teacher strike[d] | 3,263 (14.8) | 97 (12.6) | 653 (14.5) | 0 (0.0) | 21 (1.4) | 61 (2.4) | 9 (0.7) | 2,239 (31.0) | 183 (5.6) |
| Environmental problem | 203 (0.9) | 4 (0.5) | 10 (0.2) | 92 (9.2) | 3 (0.2) | 14 (0.6) | 3 (0.2) | 29 (0.4) | 48 (1.5) |
| Building/utility problem | 113 (0.5) | 33 (4.3) | 6 (0.1) | 9 (0.9) | 17 (1.1) | 12 (0.5) | 12 (0.9) | 10 (0.1) | 14 (0.4) |
| Illness | 229 (1.0) | 0 (0.0) | 0 (0.0) | 3 (0.3) | 2 (0.1) | 1 (0.0) | 26 (1.9) | 85 (1.2) | 112 (3.5) |
| Violence | 38 (0.2) | 0 (0.0) | 0 (0.0) | 7 (0.7) | 25 (1.7) | 4 (0.2) | 0 (0.0) | 1 (0.0) | 1 (0.0) |
| Season, n (column %) | | | | | | | | | |
| Fall (Sep- Nov) | 10,220 (46.2) | 571 (74.1) | 4,064 (90.1) | 30 (3.0) | 129 (8.6) | 417 (16.7) | 890 (64.4) | 2,753 (38.2) | 1,366 (42.1) |
| Winter (Dec-Feb) | 7,490 (33.9) | 30 (3.9) | 195 (4.3) | 892 (89.6) | 1,331 (88.9) | 1,745 (69.9) | 218 (15.8) | 1,398 (19.4) | 1,681 (51.8) |
| Spring (Mar-May) | 2,241 (10.1) | 158 (20.5) | 10 (0.2) | 69 (6.9) | 14 (0.9) | 330 (13.2) | 40 (2.9) | 1,579 (21.9) | 41 (1.3) |
| Summer (Jun-Aug) | 2,161 (9.8) | 11 (1.4) | 244 (5.4) | 5 (0.5) | 24 (1.6) | 3 (0.1) | 234 (16.9) | 1,485 (20.6) | 155 (4.8) |
| Urbanicity, n (column %) | | | | | | | | | |
| City | 7,682 (34.7) | 277 (36.0) | 2,489 (55.2) | 310 (31.1) | 343 (22.9) | 474 (19.0) | 352 (25.5) | 2,355 (32.6) | 1,082 (33.4) |
| Suburban | 7,402 (33.5) | 313 (40.7) | 1,753 (38.8) | 139 (14.0) | 278 (18.6) | 1,208 (48.4) | 265 (19.2) | 2,869 (39.8) | 577 (17.8) |
| Town | 2,089 (9.5) | 39 (5.1) | 57 (1.3) | 192 (19.3) | 278 (18.6) | 201 (8.1) | 194 (14.0) | 636 (8.8) | 492 (15.2) |
| Rural | 4,877 (22.1) | 141 (18.3) | 197 (4.4) | 353 (35.4) | 595 (39.7) | 608 (24.4) | 567 (41.0) | 1,336 (18.5) | 1,080 (33.3) |
| Not specified | 62 (0.3) | 0 (0.0) | 17 (0.4) | 2 (0.2) | 4 (0.3) | 4 (0.2) | 4 (0.3) | 19 (0.3) | 12 (0.4) |
| HHS region[e], n (column %) | | | | | | | | | |

*(Continued)*

**Table 1.** (Continued)

| | | PUSC by Academic Year | | | | | | | |
|---|---|---|---|---|---|---|---|---|---|
| | **Total** | **2011–12** | **2012–13** | **2013–14** | **2014–15** | **2015–16** | **2016–17** | **2017–18** | **2018–19** |
| HHS 1 | 1,021 (4.6) | 475 (61.7) | 459 (10.2) | 1 (0.1) | 46 (3.1) | 0 (0.0) | 2 (0.1) | 34 (0.5) | 4 (0.1) |
| HHS 2 | 3,150 (14.3) | 30 (3.9) | 3,022 (67.0) | 1 (0.1) | 83 (5.5) | 1 (0.0) | 1 (0.1) | 2 (0.0) | 10 (0.3) |
| HHS 3 | 3,109 (14.1) | 2 (0.3) | 85 (1.9) | 138 (13.9) | 416 (27.8) | 1,418 (56.8) | 20 (1.5) | 835 (11.6) | 195 (6.0) |
| HHS 4 | 6,944 (31.4) | 6 (0.8) | 24 (0.5) | 294 (29.5) | 816 (54.5) | 583 (23.4) | 920 (66.6) | 2,765 (38.3) | 1,536 (47.4) |
| HHS 5 | 1,643 (7.4) | 2 (0.3) | 657 (14.6) | 188 (18.9) | 42 (2.8) | 22 (0.9) | 0 (0.0) | 132 (1.8) | 600 (18.5) |
| HHS 6 | 3,163 (14.3) | 0 (0.0) | 255 (5.7) | 190 (19.1) | 10 (0.7) | 329 (13.2) | 220 (15.9) | 2,145 (29.7) | 14 (0.4) |
| HHS 7 | 418 (1.9) | 1 (0.1) | 4 (0.1) | 150 (15.1) | 71 (4.7) | 0 (0.0) | 64 (4.6) | 0 (0.0) | 128 (4.0) |
| HHS 8 | 20 (0.1) | 0 (0.0) | 0 (0.0) | 12 (1.2) | 3 (0.2) | 0 (0.0) | 3 (0.2) | 0 (0.0) | 2 (0.1) |
| HHS 9 | 1,614 (7.3) | 1 (0.1) | 0 (0.0) | 1 (0.1) | 10 (0.7) | 45 (1.8) | 66 (4.8) | 1,298 (18.0) | 193 (6.0) |
| HHS 10 | 1,030 (4.7) | 253 (32.9) | 7 (0.2) | 21 (2.1) | 1 (0.1) | 97 (3.9) | 86 (6.2) | 4 (0.1) | 561 (17.3) |
| **Unit of analysis: Teacher** | | | | | | | | | |
| Number of teachers affected by PUSCs[f], n (row %) | 846,473 | 25,996 (3.1) | 184,470 (21.8) | 32,319 (3.8) | 50,017 (5.9) | 101,873 (12.0) | 50,935 (6.0) | 297,031 (35.1) | 103,832 (12.3) |
| **Unit of analysis: Student** | | | | | | | | | |
| Number of students affected by PUSCs[g], n (row %) | 13,403,555 | 385,559 (2.9) | 2,724,961 (20.3) | 497,907 (3.7) | 765,508 (5.7) | 1,578,493 (11.8) | 790,268 (5.9) | 4,953,889 (37.0) | 1,706,970 (12.7) |
| Percent of students eligible for free/reduced lunch[h], median (IQR) | 62.9 (39.9–84.5) | 46.3 (15.9–71.2) | 71.8 (24.2–88.7) | 62.3 (47.2–77.2) | 54.8 (39.3–71.6) | 57.8 (35.6–79.3) | 79.1 (54.2–99.3) | 62.1 (43.1–80.5) | 66.6 (43.5–92.9) |
| **Unit of analysis: Student-day** | | | | | | | | | |
| Number of student-days lost[i] (row %) | 91,565,170 | 2,285,301 (2.5) | 16,539,006 (18.1) | 2,693,635 (2.9) | 5,359,471 (5.9) | 8,755,956 (9.6) | 5,871,793 (6.4) | 37,159,793 (40.6) | 12,900,215 (14.1) |
| By cause of PUSC (column %) | | | | | | | | | |
| Weather | 24,870,455 (27.2) | 1,871,769 (81.9) | 1,230,028 (7.4) | 2,265,328 (84.1) | 4,885,901 (91.2) | 7,275,115 (83.1) | 1,905,088 (32.4) | 1,635,202 (4.4) | 3,802,024 (29.5) |
| Natural disaster | 51,056,023 (55.8) | 44,212 (1.9) | 12,462,395 (75.4) | 83,240 (3.1) | 37,602 (0.7) | 1,044,498 (11.9) | 3,788,014 (64.5) | 25,729,051 (69.2) | 7,867,011 (61.0) |
| Budget/teacher strike[d] | 14,125,753 (15.4) | 319,993 (14.0) | 2,797,242 (16.9) | 0 (0.0) | 334,560 (6.2) | 381,711 (4.4) | 79,965 (1.4) | 9,412,326 (25.3) | 799,956 (6.2) |
| Environmental problem | 568,964 (0.6) | 8,595 (0.4) | 34,553 (0.2) | 286,227 (10.6) | 12,960 (0.2) | 18,565 (0.2) | 4,685 (0.1) | 66,440 (0.2) | 136,939 (1.1) |
| Building/utilities problem | 269,595 (0.3) | 40,732 (1.8) | 14,788 (0.1) | 23,320 (0.9) | 20,087 (0.4) | 31,827 (0.4) | 46,402 (0.8) | 32,762 (0.1) | 59,677 (0.5) |
| Illness | 547,597 (0.6) | 0 (0.0) | 0 (0.0) | 2,440 (0.1) | 6,066 (0.1) | 0 (0.0) | 47,639 (0.8) | 257,372 (0.7) | 234,080 (1.8) |
| Violence | 126,783 (0.1) | 0 (0.0) | 0 (0.0) | 33,080 (1.2) | 62,295 (1.2) | 4,240 (0.1) | 0 (0.0) | 26,640 (0.1) | 528 (0.0) |

[a] PUSC is defined as a school closure lasting ≥5 school days, excluding any scheduled days off.

[b] Percentages may not add up to 100%, as they are rounded to the nearest tenth of a percent.

[c] Schools were counted once for each time they experienced a PUSC across the study period.

[d] All but one budget-related PUSC were due to teacher strike, the remaining was attributed to a funding issue.

[e] Regions of the United States Department of Health & Human Services (HHS). https://www.hhs.gov/about/agencies/regional-offices/index.html

[f] Teachers were counted once for each PUSC. Part-time teaching positions were reported as a fraction of one full-time position. Information missing for 410 schools.

[g] Students were counted once for each PUSC. Information missing for 77 schools.

[h] Data available for public schools only, among which information was missing for 564 schools.

[i] Student-days lost is estimated as the summation of the number of students per school closed multiplied by the duration of closure. Information on total students missing for 77 schools.

represented individual school closures. The 22,112 PUSCs occurred in 19,582 unique schools (S1 Table). A total of 17,462 (89.2%) schools had a single PUSC, while the remaining 2,120 (10.8%) schools had multiple PUSCs (ranging from two to four PUSCs) over the study period (S2 Table). The largest number of schools affected by multiple PUSCs across the study period were in rural and suburban areas (39.1% and 27.8%, respectively) than schools in cities (16.6%) and towns (16.5%). Schools in HHS regions 1, 3, 4, and 6 experienced the largest number of multiple PUSCs (7.6%, 25.8%, 42.5%, and 13.1%, respectively) compared to schools in the other HHS regions (S2 Table).

During the eight-year period, more than 13 million students and 800,000 teachers were affected by PUSC (Table 1). Among public-school PUSCs, a median of 62.9% of students were eligible for subsidized school meals. The median duration of PUSCs was 6 days (5th, 95th percentile: 5, 11). The majority of PUSCs were in cities (34.7%) and suburban areas (33.5%), as compared to rural areas (22.1%) or towns (9.5%). Elementary schools accounted for the largest proportion of PUSCs (41.6%), followed by elementary-middle schools (19.9%), and high schools (16.4%) (Table 1). The cumulative incidence of total PUSCs per 100 schools was 24.2 for middle schools, compared to 22.9 for elementary schools and 20.5 for high schools (S1 Table).

Nearly half of all PUSCs were due to natural disasters (47.5%); the majority of these occurred during the 2012–2013 and 2017–2018 academic years (Table 1). Weather-related PUSCs accounted for 35.1% of PUSCs. Budget/teacher strike caused 14.8% of the PUSCs, the majority of which were reported during the 2017–2018 academic year. Illness, environmental issue, building/utility problem, and violence accounted for a combined 2.6% of all PUSCs. Violence-related PUSCs were highest in 2014–2015, among which 96% were the result of a district-wide closures in response to local unrest and the perceived threat to student safety. Actualized violence in schools (e.g., shootings, stabbings) were documented to cause PUSC in three school years (S3 Table).

The highest proportion of PUSCs were recorded during the fall season (46.2%), followed by winter (33.9%) (Table 1). Most of the PUSCs during the fall season were due to natural disasters (83.4%) (Fig 1). In contrast, most PUSCs during the winter season were weather-related (80.9%). Budget/teacher strikes were the most common cause of PUSCs during the spring (71.6%).

The distribution of PUSCs varied across the United States, with the largest number of PUSCs occurring in HHS Region 4 (31.4%), followed by Region 6 (14.3%), Region 2 (14.3%), and Region 3 (14.1%) (Table 1). Natural-disaster related PUSCs were clustered in Region 2 during the 2012–2013 academic year, and in Region 4 from 2015 to 2019 (S4 Table). Wildfires were most common during two consecutive academic years, 2017–2018 and 2018–2019, with nearly all occurring in California (S3 and S5 Tables). Weather-related PUSCs occurred in all but one HHS region (Region 8) (S4 Table), with a higher number of closures during the 2014–2015, 2015–2016, and 2018–2019 academic years (Table 1, S3 Table). The seasonal distribution of weather-related PUSCs varied by region. Winter was the predominant season for weather-related PUSCs both nationally and separately in HHS Regions 3, 4, 5, 7, and 10; representing the Mid-Atlantic, Southeast, Great Lakes, Central Plains, and Pacific Northwest regions (S6 Table). Aside from winter, the fall season held the majority of weather-related PUSCs in HHS Regions 1, 2, and 9, encompassing the Northeast and Pacific Southwest; and HHS Region 6, the South Central states, had the greatest numbers of weather-related PUSCs in summer and spring.

Among illness-related PUSCs, the majority were recorded in Kentucky and Tennessee (52.4% and 34.1%, respectively) (S7 Table), both in HHS Region 4, during the eight-year

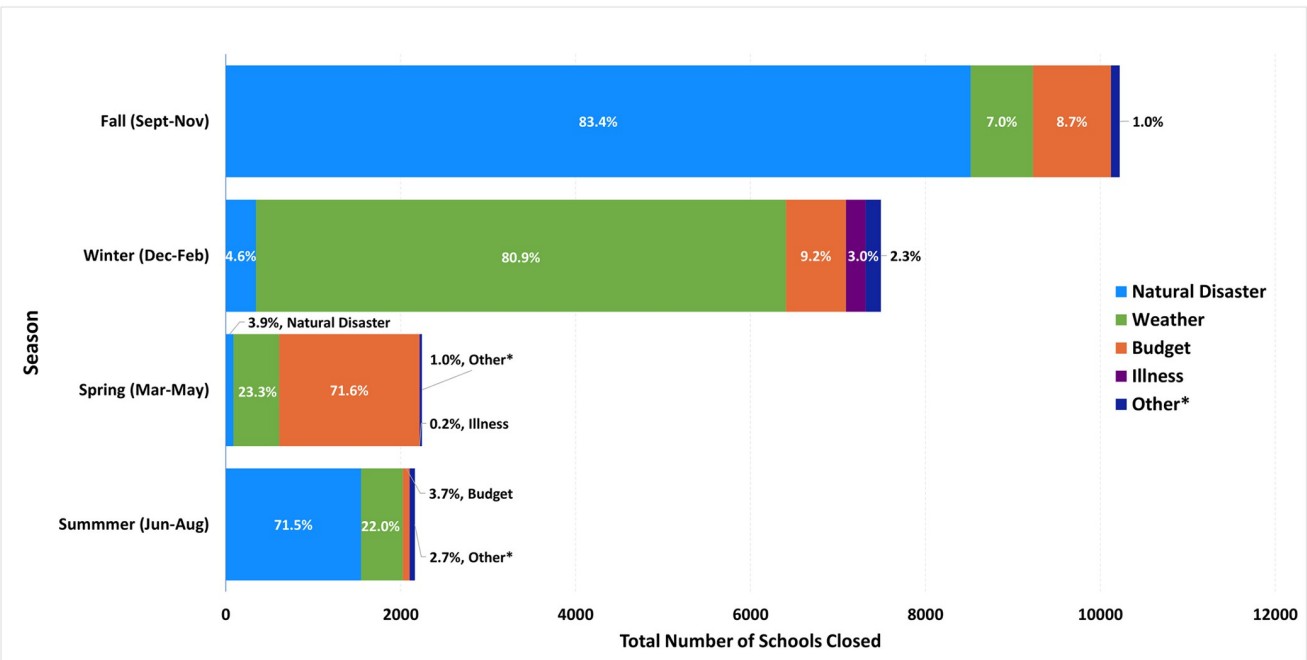

**Fig 1. Reasons for prolonged unplanned school closures (PUSCs)[a,b], by season, United States, 2011–2019[†]. [a]** PUSC defined as a school closure lasting ≥5 school days, excluding any scheduled days off. **[b]** Percentages may not add up to 100%, as they are rounded to the nearest tenth of a percent. *Other reasons are environmental problem, building/utility problem, and violence.

period. The illness-related PUSCs were primarily attributed to influenza and influenza-like illnesses (S7 Table).

Twenty specific reasons accounted for the majority (74,049,042 or 80.9%) of student-days lost during the study period; sixteen of these individually caused a loss of >1 million student-days (Table 2). Eight of the top twenty specific reasons were associated with hurricanes or named storms, and additional seven were due to weather events; hence, natural disasters and extreme weather were associated with sixteen of the top twenty reasons. Hurricanes Irma and Harvey, which occurred in 2017, and Hurricane Sandy in 2012, caused more than 10 million student-days lost each. Large snowfalls in Maryland (2016), Tennessee (2015), North Carolina (2018), and Washington (2019) also caused substantial numbers of student-days lost. Just four PUSC among the top twenty (by the number of student-days lost) were associated with reasons other than natural disasters or extreme weather; all four were attributed to teacher strikes [one each in Illinois (2012), and in Arizona, Oklahoma, and West Virginia (2018)] and led to a combined total of approximately 12.1 million student-days lost (Table 2).

With regard to the geographic distribution of the PUSCs during the study period, they were identified in all but one state (Wyoming) and DC. West Virginia was the most affected state in terms of student-days lost per 1,000 enrolled students (averaging more than 1,600 days lost per 1,000 students per year), where most of the closures were due to teacher strikes and severe weather (Figs 2 and 3). Trailing West Virginia, annual student-days lost per 1,000 students averaged between 500–999 in 6 states (CT, FL, MD, NC, OK, SC) and 100–499 in 14 states (AK, AZ, GA, IL, IN, KY, LA, NJ, NY, OR, TN, TX, VA, WA), with all remaining states averaging fewer student-days lost per year.

The association between selected school and PUSC characteristics and the PUSC lasting ≥6 days, are shown in Table 3. Compared to weather-related PUSCs, closures due to budget/

**Table 2. Top twenty reasons for prolonged unplanned school closures (PUSCs) according to the number of student-days lost, United States, 2011–2019[a].**

| Category | Specific reason | Date of PUSC | States affected by PUSC | # of schools with PUSCs | Estimated number of student-days lost[b] |
|---|---|---|---|---|---|
| Natural disaster | Hurricane Irma | Sep 2017 | 2 (FL, GA) | 2,456 | 13,157,768 |
| Natural disaster | Hurricane Sandy | Oct 2012 | 7 (CT, MD, NC, NJ, NY, PA, WV) | 3,407 | 12,453,449 |
| Natural disaster | Hurricane Harvey | Aug 2017 | 1 (TX) | 1,480 | 11,061,696 |
| Natural disaster | Hurricane Florence | Sep 2018 | 3 (NC, SC, VA) | 872 | 5,492,147 |
| Budget/teacher strike | Teacher strike | Apr 2018 | 1 (AZ) | 910 | 4,116,332 |
| Weather | Snow | Jan 2016 | 1 (MD) | 1113 | 3,925,600 |
| Natural disaster | Hurricane Matthew | Oct 2016 | 4 (FL, GA, NC, SC) | 844 | 3,417,971 |
| Budget/teacher strike | Teacher strike | Mar-Apr 2018 | 1 (OK) | 654 | 3,312,029 |
| Budget/teacher strike | Teacher strike | Sep-Oct 2012 | 1 (IL) | 649 | 2,793,390 |
| Weather | Snow | Feb 2015 | 1 (TN) | 536 | 2,290,532 |
| Budget/teacher strike | Teacher strike | Feb 2018 | 1 (WV) | 671 | 1,916,117 |
| Weather | Storm | Aug 2016 | 1 (LA) | 211 | 1,525,712 |
| Weather | Storm | Apr 2016 | 1 (TX) | 250 | 1,453,555 |
| Weather | Nor'easter | Oct 2011 | 4 (CT, MA, NH, NJ) | 498 | 1,448,828 |
| Natural disaster | Hurricane Michael | Oct 2018 | 4 (AL, FL, GA, VA) | 222 | 1,258,515 |
| Weather | Snow | Dec 2018 | 1 (NC) | 335 | 1,006,335 |
| Natural disaster | Hurricane Joaquin | Oct 2015 | 1 (SC) | 267 | 970,645 |
| Weather | Snow | Feb 2019 | 1 (WA) | 296 | 868,745 |
| Natural disaster | Wildfire | Oct 2017 | 1 (CA) | 236 | 851,485 |
| Natural disaster | Tropical Storm Isaac | Aug 2012 | 2 (LA, MS) | 244 | 728,191 |

[a] PUSC defined as a school closure lasting ≥5 school days, excluding any scheduled days off.

[b] Student-days lost was estimated by multiplying the number of students per school by the duration of closure. Number of students per school closed was obtained from the National Center for Education Statistics. Information on total number of students was not available for 58 schools.

AL = Alabama, AZ = Arizona, CA = California, CT = Connecticut, FL = Florida, GA = Georgia, IL = Illinois, LA = Louisiana, MA = Massachusetts, MD = Maryland, MS = Mississippi, NC = North Carolina, NH = New Hampshire, NJ = New Jersey, NY = New York, OK = Oklahoma, PA = Pennsylvania, SC = South Carolina, TN = Tennessee, TX = Texas, VA = Virginia, WA = Washington, WV = West Virginia.

teacher strikes (aOR: 27.8, 95% CI: 23.9–32.3), natural disasters (aOR: 3.5, 95% CI: 3.3–3.7), and building/utility issues (aOR: 2.8, 95% CI: 1.8–4.3) were more likely to last ≥6 days, whereas violence-related PUSCs were less likely to result in ≥6 closure days (aOR: 0.1, 95% CI: 0.0–0.4). District-wide PUSCs were more likely to last ≥6 days (aOR: 1.8, 95% CI: 1.3–2.4) when compared with individual school-level PUSCs. Schools with higher poverty levels (≥25% of students eligible for subsidized school meals) were more likely to have closures lasting ≥6 days.

## Discussion

To our knowledge, the present study is the first to explore the causes, characteristics, and patterns of PUSCs in the United States due to any cause and outside of a global pandemic. Our results demonstrate that prolonged unplanned school closures lasting an entire school week or longer occur annually in the United States due to a variety of causes and are associated with a substantive loss of student-days for in-school learning. A substantial proportion of students in

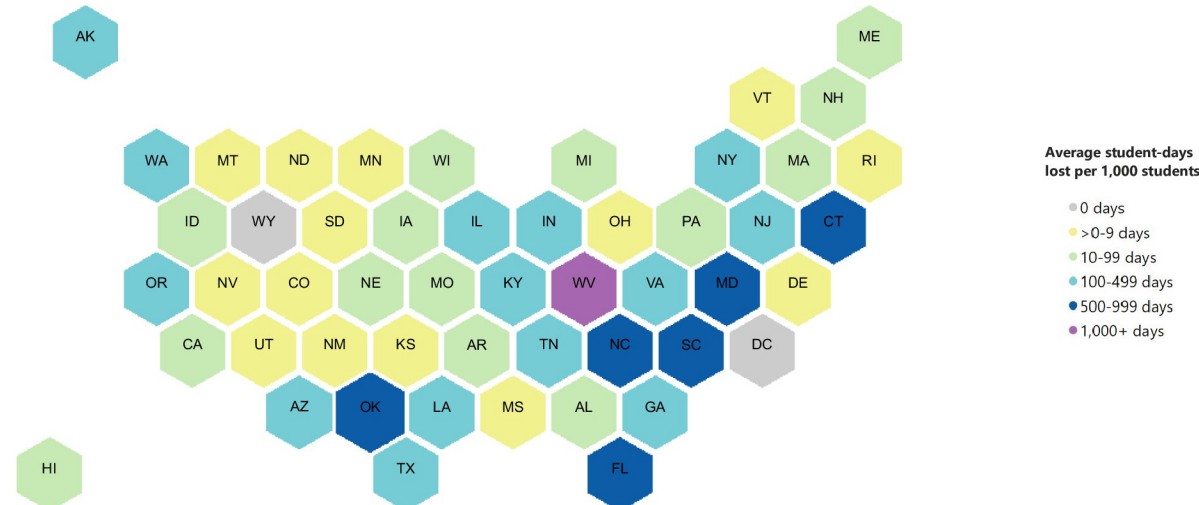

**Fig 2. Student-days lost to prolonged unplanned school closures per 1,000 students per year, by state, United States, 2011–2019[a].** [a] PUSC defined as a school closure lasting ≥5 school days, excluding any scheduled days off. Student-days lost per 1,000 students per year = ((average student-days lost per year due to PUSCs) / (average number of students enrolled per year in all K-12 schools)) x 1000. Student enrollment data was obtained from the National Center for Education Statistics for all schools, i.e., schools with closures of 0 days, 1–4 days, or ≥5 days. Enrollment data was not available for 77 schools that experienced PUSC.

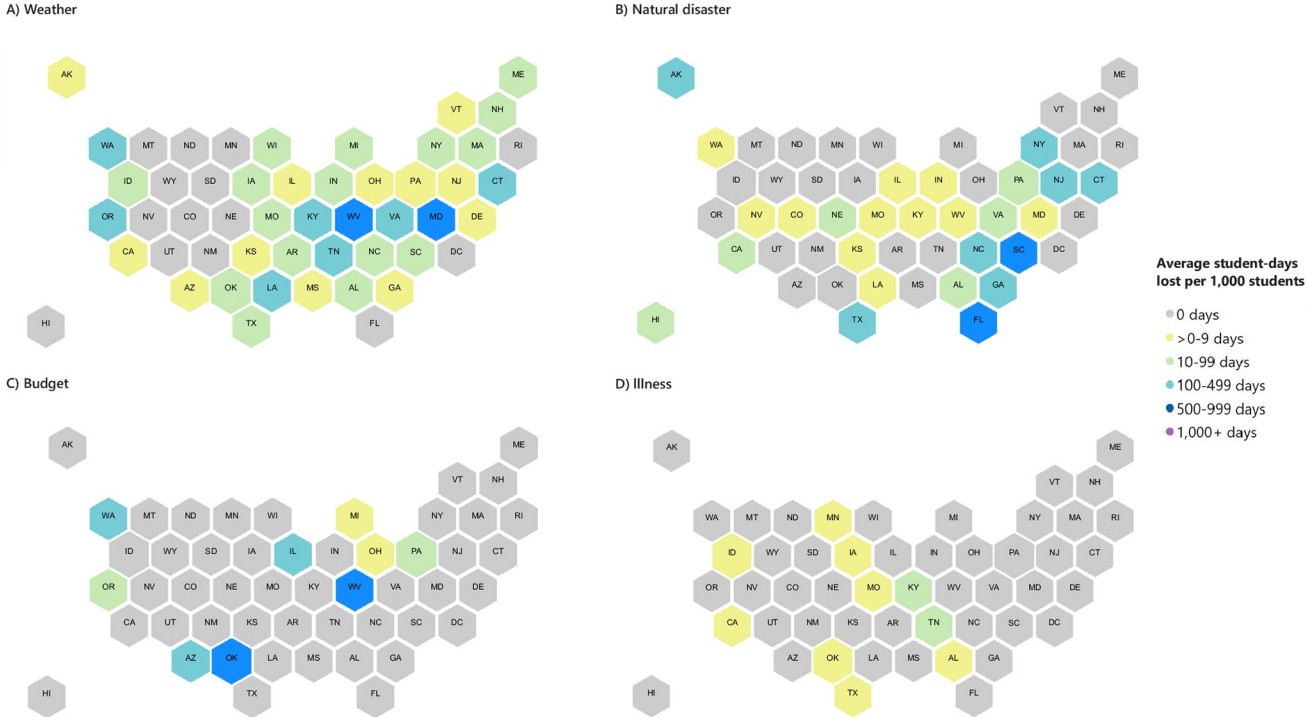

**Fig 3. Student-days lost to prolonged unplanned school closures per 1,000 students per year, by state and cause, United States, 2011–2019[a].** (A) Weather; (B) Natural disasters; (C) Budget; (D) Illness. [a] PUSC defined as a school closure lasting ≥5 school days, excluding any scheduled days off. Student-days lost per 1,000 students per year = ((average student-days lost per year due to PUSCs) / (average number of students enrolled per year in all K-12 schools)) x 1000. Student enrollment data was obtained from the National Center for Education Statistics for all schools, i.e., schools with closures of 0 days, 1–4 days, or ≥5 days. Enrollment data was not available for 77 schools that experienced PUSC. [b] All but one budget-related PUSC were due to teacher strike, the remaining was attributed to a funding issue.

**Table 3. Selected characteristics associated with school closures of ≥6 days among schools with PUSCs, United States, 2011–2019 [a].**

| Characteristics | Proportion of schools that closed for ≥6 days (%)[b] | Unadjusted odds ratio (95% CI)[c] | Adjusted odds ratio (95% CI)[c] |
|---|---|---|---|
| **Reason for closure** | | | |
| Weather | 2,874/7,770 (37.0) | 1.0 (referent) | 1.0 (referent) |
| Budget/teacher strike[d] | 3,049/3,264 (93.4) | 24.3 (21.0–28.1) | 27.8 (23.9–32.3) |
| Natural disaster | 6,968/10,496 (66.4) | 3.4 (3.2–3.6) | 3.5 (3.3–3.7) |
| Building/utility problem | 65/113 (57.5) | 2.3 (1.6–3.4) | 2.8 (1.8–4.3) |
| Environmental problem | 119/203 (58.6) | 2.4 (1.8–3.2) | 1.3 (0.9–1.9) |
| Illness | 85/229 (37.1) | 1.0 (0.8–1.3) | 1.1 (0.8–1.5) |
| Violence | 3/38 (7.9) | 0.1 (0.0–0.5) | 0.1 (0.0–0.4) |
| **Type of closure** | | | |
| Individual school | 299/527 (56.7) | 1.0 (referent) | 1.0 (referent) |
| District-wide | 12,864/21,585 (59.6) | 1.1 (0.9–1.3) | 1.8 (1.3–2.4) |
| **Urbanicity[e]** | | | |
| Town | 1,163/2,089 (55.7) | 1.0 (referent) | 1.0 (referent) |
| Suburban | 4,770/7,402 (64.4) | 1.4 (1.3–1.6) | 1.5 (1.3–1.7) |
| City | 4,412/7,682 (57.4) | 1.0 (1.0–1.2) | 0.8 (0.7–0.9) |
| Rural | 2,774/4,877 (56.9) | 1.0 (0.9–1.2) | 1.2 (1.1–1.4) |
| **Percent of students eligible for subsidized school meals[f]** | | | |
| <25% | 1,705/2,999 (56.9) | 1.0 (referent) | 1.0 (referent) |
| 25–49% | 2,616/4,420 (59.2) | 1.1 (1.0–1.2) | 1.3 (1.2–1.5) |
| 50–74% | 3,684/6,103 (60.4) | 1.2 (1.0–1.3) | 1.4 (1.2–1.5) |
| >75% | 4,640/7,698 (60.3) | 1.2 (1.0–1.3) | 1.2 (1.1–1.4) |
| **HHS region[g]** | | | |
| HHS 8 | 1/19 (5.0) | 1.0 (referent) | - |
| HHS 1 | 381/1021 (37.3) | 11.3 (1.5–84.7) | - |
| HHS 2 | 1,128/3,150 (35.8) | 10.6 (1.4–79.1) | - |
| HHS 3 | 1,929/3,109 (62.1) | 31.0 (4.2–231.9) | - |
| HHS 4 | 4,823/6,944 (69.5) | 43.2 (5.8–322.3) | - |
| HHS 5 | 803/1,643 (48.9) | 18.1 (2.4–135.7) | - |
| HHS 6 | 2,353/3,163 (74.4) | 55.1 (7.4–412.1) | - |
| HHS 7 | 70/418 (16.8) | 3.8 (0.5–28.9) | - |
| HHS 9 | 1,243/1,614 (77.0) | 63.6 (8.5–476.2) | - |
| HHS 10 | 432/1030 (42.9) | 13.7 (1.8–102.7) | - |

[a] PUSC defined as a school closure lasting ≥5 school days, excluding any scheduled days off.

[b] The median duration of PUSC is 6 days. Numerator is the number of schools that closed for ≥6 days; denominator is the number of schools that closed for ≥5 days.

[c] Unadjusted and adjusted odds ratios were computed using univariate and multivariable logistic regression models, respectively. The independent variables are reason for closure, type of closure, urbanicity, and percent of students eligible for subsidized meals. The dependent variable is duration of closure (0 = 5 days; 1 = ≥6 days). HHS region was excluded from the multivariable model because of collinearity with reason for closure.

[d] All but one budget-related PUSC were due to teacher strike, the remaining was attributed to a funding issue.

[e] Urbanicity was not specified for 62 schools.

[f] Data only for public schools, and among them 564 have missing value. Both the univariate and multivariable logistic regression models exclude the 328 private schools.

[g] Regions of the United States Department of Health & Human Services (HHS). https://www.hhs.gov/about/agencies/regional-offices/index.html

the affected schools were eligible for subsidized school meals. The majority of PUSCs were due to natural disasters and severe weather, affecting most states. Teacher strikes were the leading non-weather/natural disaster-related cause of prolonged closures.

Our results should be considered in context of at least five limitations. First, the data are limited to closures that were found in publicly available online sources through daily systematic searches and therefore may not be complete or entirely accurate, particularly missing those closures announced through social media [6, 7] and those announced through non-public methods (email, text message, etc.). Second, while HHS Regions 2 and 9 include US territories, the data are limited to the 50 US states and DC and therefore US territories are not represented in these data. Third, the data are limited to information found in English language sources. Fourth, the data on PUSCs are limited to closures for which reopening dates could be found. Information regarding school reopening may have been sent directly to the school community via texts, emails, or phone calls, thereby bypassing the news media. Fifth, particularly in the case of disrupted infrastructures and facilities, schools and districts may not have been able to report updates on school status to the public media. These limitations could have led to an underestimation of prolonged closures, particularly for private, small, or remotely situated schools for which public news sources may be fewer. However, one notable advantage of this data collection method is that it enabled long-term evaluation of school closure trends without imposing any reporting burden on schools and school districts.

We previously reported on the methods and rationale for this data collection, along with the initial two years of data on closures of any length [3]. While the previous study considered closures of ≥4 days prolonged-duration closures, we chose to evaluate closures lasting ≥5 days in the present study to both emulate the length of a natural school week and because a school week is a unit of planning for possible use of preemptive school closures as one of the key community-level nonpharmaceutical interventions (NPIs) that may be recommended during severe influenza pandemics [8]. The previous study reported that 3.7% of all closure events had a duration of ≥4 days. Using a cut-off of ≥5 days, we found that 1.9% of all closure events were prolonged.

## PUSCs in context

**Natural disasters and severe weather.**   Our study shows that natural disaster-related events led to PUSCs in more than half of US states, and were attributed to major hurricanes, wildfires, floods, earthquakes, tornadoes, and volcanoes. These events often impacted broad geographic areas, sometimes multiple states. Hurricane Sandy in 2012, for instance, affected the entire eastern seaboard of the US and led to PUSCs in seven states. This included the closure of all New York City public schools, serving more than 1 million students, for an entire week; with many schools remaining closed into the next week [9]. Similarly, due to the effects of Hurricanes Harvey and Irma in 2017, an estimated 8.5 million K-12 students missed school days in seven states [10]. Hurricane Harvey alone led to more than 3,000 schools closed in four states for durations ranging from 1–19 days [11]; all resulting PUSCs were in Texas, where 20% of public schools were affected and as many as 1.4 million students remained out of schools for at least a week [12]. These events may also have led to the complete destruction of school facilities, and even permanent closure [13–15]. In contrast, some documented PUSCs were undertaken in preparation for hurricane landings that have not led to the physical destruction of schools and communities; however, as seen in an example associated with the 2012 Hurricane Isaac, communities had to find ways to cope with the resulting PUSCs [16].

Like natural disaster-related events, weather-related PUSCs are often attributed to the weather event itself as well as the lasting effects of the event. While winter weather events (snow, ice, extreme cold) may be more common and severe in northern states, winter weather PUSCs are often implemented out of a concern for the safety of students and teachers while traveling to school [17]. While southern states may experience less frequent and severe winters, they are also less likely to possess the equipment and supplies necessary to repair damaged or impeded infrastructure so that students and teachers can safely travel to school [18]. In addition to the lingering effects of a storm such as snowfall, flooding, or ice, other secondary consequences of these events such as damage to property, infrastructure, or utilities impact school operations. These post-event consequences of natural disasters and severe weather further delay the return to normal school operations [10, 11, 15].

**Other causes.** Numerous other causes such as teacher strikes, illness, environmental issues, building/utilities issues, and violence also led to PUSCs during the 8-year study period. Illness-related closures were most frequently associated with influenza or influenza-like illness; detailed reporting on these closures regardless of duration is provided elsewhere [19]. As measured by the number of student-days lost, closures due to teacher strikes had a substantial impact. Teacher strike-related PUSCs generally occurred as full district-closures, as in Chicago in 2012 [20], or larger state closures affecting multiple districts across a state or region, such as Oklahoma and West Virginia in 2018 [21, 22]. School shootings can lead to prolonged closures because of the time needed by police to finish collecting evidence from the crime scene and for school workers to make repairs for removing visible signs of the violence (e.g., blood stains, bullet holes) [23].

## Distribution of the PUSC burden in the eve of the COVID-19 pandemic

Unlike the effectively nationwide PUSCs during the spring of 2020 (March-June) in response to the COVID-19 pandemic in the United States, the burden of PUSCs across the study period was distributed unequally with regard to geography, seasonality, and populations affected. The highest number of PUSCs during the study period occurred in HHS Regions 2, 3, 4, and 6, encompassing the densely populated states of New York, New Jersey, as well as the Southern United States that are frequently in the path of hurricanes and tropical storms (Alabama, Florida, Georgia, Louisiana, Mississippi, North Carolina, South Carolina, and Texas). The majority of PUSCs in these regions were attributed to a handful of natural disasters during the Fall season: Hurricane Sandy (2012), Hurricane Irma (2017), Hurricane Harvey (2017), Hurricane Florence (2018), and Hurricane Matthew (2016). There were a substantial number of disadvantaged students in schools with PUSCs, as indicated by the proportion of students eligible for subsidized school meals in these schools.

## Costs and consequences of PUSCs

The costs and consequences of PUSC are topics of intense contemporary research interest, especially in view of COVID-19-related school closures, as well as subsequent modifications of instruction delivery methods and implementation of alternate methods to continue subsidized school meal programs [24]. This study illustrates that even prior to the COVID-19 pandemic, anywhere from several hundred to several thousand schools were closed for a prolonged period (one entire school week or longer) each year. Because PUSCs will likely continue to be necessitated by future natural disasters, pandemics, or other unforeseen circumstances, it is important to anticipate and prepare ahead of these events for the disruption of routine in-school education and other services, most notably school-based supplemental feeding programs for both students and their families, as well as staff and administrators. Among studies

that have identified costs and consequences of PUSCs, uncertainty of duration, inability to arrange alternative childcare, lost income, and missing free/reduced priced meals were often reported [16, 20, 25]. Disruption to normal methods of education delivery may lead to negative impacts on student learning (i.e. "learning loss") [26, 27] and may have imposed significant challenges to school practitioners in the transition to distance learning, particularly among schools that had not implemented distance learning prior to the event [28]. Meanwhile, the disruption in the provision of auxiliary services, such as school-based subsidized meal programs, may result in decreased food security [29]. With parents unable to stay home, PUSC may also lead to increases in self-care among students [30], which is associated with elevated risk to physical and emotional health [31]. As previously described during a CDC stakeholder meeting in 2008, the consequences of PUSC may also disproportionately impact those of racial/ethnic minority populations, including greater gaps in access to meals, educational resources, and childcare [32].

## Conclusion

In summary, our study demonstrated that while only 2% of all unplanned school closures during the 8-year study period were categorized as prolonged (lasting ≥5 days), such closures occur annually due to a variety of causes and have major impacts on schools, students, teachers, and families. The establishment of a publicly available reporting system for school closures at the state or national level could lead to a better understanding of the magnitude and duration of PUSCs, both during the inter-pandemic period like the one on which we reported here (2011–2019) and during the major public health emergencies like severe influenza outbreaks [19] or during the respiratory infectious disease pandemics [8, 24, 33]. Further studies are needed to elucidate the economic and social consequences of PUSCs, both those occurring outside of a pandemic and those which are pandemic-related, particularly for those populations that are disproportionately affected, and inform preparedness guidelines for future occurrences of prolonged unplanned school closures. The numerous costs and consequences of these closures should be anticipated and ameliorated ahead of the crises by ensuring the continued availability of quality education through distance learning and advanced planning for the continued provision of auxiliary services. Contemporary research in the education sector is already starting to point out that the crisis-level emergency remote teaching must not be confused with well-designed online learning programs [34]. Innovative online learning initiatives implemented during the COVID-19 pandemic show promising strategies for providing a high-quality learning experience for students [35].

Given the results of this study which documented that prolonged school closures occur annually even outside of a global pandemic, and given the experience of the still-ongoing COVID-19 pandemic [24], it appears that there is already a solid investment case for creating sustainable solutions for high quality digital learning and innovative supplemental feeding programs nationwide before next disaster strikes, with particular attention and urgency for disaster-prone areas and socio-economically vulnerable communities.

## Supporting information

**S1 Table. Cumulative incidence of prolonged unplanned school closure (PUSC) per 100 schools, United States, 2011–2019[a].** [a] PUSC is defined as a school closure lasting ≥5 school days, excluding any scheduled days off. Cumulative incidence of PUSC per 100 schools was computed by dividing the number of schools with PUSCs during the study period (2011–2019) by the total number of all K-12 schools and then multiplying by 100. [b] Schools were counted each time they experienced a PUSC across the 8-year study period. [c] Schools with

multiple PUSCs were counted only once across the 8-year study period. [d] The total number of K-12 schools was obtained from the National Center for Education Statistics (NCES) by summing the number of schools reported for each academic year (2011–2012 through 2018–2019) and dividing by eight. [e] Grade span was not specified for 189 PUSCs, across 186 unique schools. [f] Urbanicity was not specified for 62 schools. [g] Regions of the United States Department of Health & Human Services (HHS). https://www.hhs.gov/about/agencies/regional-offices/index.html.
(DOCX)

**S2 Table. Unique schools by the number of prolonged unplanned school closure (PUSC) events experienced, United States, 2011–2019[a,b].** [a] PUSC is defined as a school closure lasting ≥5 school days, excluding any scheduled days off. [b] Percentages may not add up to 100%, as they are rounded to the nearest tenth of a percent. [c] Regions of the United States Department of Health & Human Services (HHS). https://www.hhs.gov/about/agencies/regional-offices/index.html. [d] Urbanicity was not specified for 5 schools with NCES IDs, and unknown for 57 schools without NCES IDs.
(DOCX)

**S3 Table. Cause subcategories of prolonged unplanned school closures (PUSCs) by academic year, United States, 2011–2019[a,b].** [a] PUSC is defined as a school closure lasting ≥5 school days, excluding any scheduled days off. [b] Percentages may not add up to 100%, as they are rounded to the nearest tenth of a percent. [c] Includes building damage from storm, fire in the building, flood from broken pipe, gas leak, unsafe building structure, vandalism/robbery, rat and roach infestation, ventilation issue. [d] Includes facility issues, no water/unsafe water, no air conditioning, no heat, no electricity.
(DOCX)

**S4 Table. Causes of prolonged unplanned school closures [a] (PUSCs) by academic year and HHS region[b], United States, 2011–2019[c].** [a] PUSC is defined as a school closure lasting ≥5 school days, excluding any scheduled days off. [b] Regions of the United States Department of Health & Human Services (HHS). https://www.hhs.gov/about/agencies/regional-offices/index.html. [c] Percentages may not add up to 100%, as they are rounded to the nearest tenth of a percent. [d] n (column percentages). [e] n (row percentages).
(DOCX)

**S5 Table. Top ten states with natural disaster-related prolonged unplanned school closures (PUSCs) by type of natural disaster, United States, 2011–2019[a,b].** [a] PUSC is defined as a school closure lasting ≥5 school days, excluding any scheduled days off. [b] An additional 17 states experienced a total of 256 (2.4%) natural disaster-related PUSCs and percentages are rounded to the nearest tenth of a percent, therefore percentages may not add up to 100%.
(DOCX)

**S6 Table. Seasonality of weather-related, prolonged unplanned school closures [a] (PUSCs) by HHS Region[b], United States, 2011–2019[c].** [a] PUSC is defined as a school closure lasting ≥5 school days, excluding any scheduled days off. [b] Regions of the United States Department of Health & Human Services (HHS). https://www.hhs.gov/about/agencies/regional-offices/index.html. [C] Percentages may not add up to 100%, as they are rounded to the nearest tenth of a percent.
(DOCX)

**S7 Table. States with illness-related prolonged unplanned school closures (PUSCs) by illness category, United States, 2011–2019[a,b].** [a] PUSC is defined as a school closure lasting ≥5

school days, excluding any scheduled days off. [b] Percentages may not add up to 100%, as they are rounded to the nearest tenth of a percent. [c] The closure announcement did not specify the type(s) of illness.
(DOCX)

## Acknowledgments

The authors would like to thank the various data collection teams over the study period, including those with Oak Ridge Associated Universities (Oak Ridge, TN), the Mayatech Corporation (Silver Spring, MD), and our unit ORISE Fellows (Ashley Jackson, Cassandra Kersten, Jeffrey Hodis, and Peter Kim). For her contributions to data visualization, we thank Sarah Moreland.

**Disclaimer**: The findings and conclusions in this report are those of the authors and do not necessarily represent the official position of the Centers for Disease Control and Prevention.

## Author Contributions

**Conceptualization:** Ferdous A. Jahan, Hongjiang Gao, Amra Uzicanin.

**Data curation:** Ferdous A. Jahan.

**Formal analysis:** Ferdous A. Jahan.

**Investigation:** Ferdous A. Jahan, Nicole Zviedrite.

**Methodology:** Hongjiang Gao, Faruque Ahmed.

**Project administration:** Nicole Zviedrite.

**Supervision:** Amra Uzicanin.

**Validation:** Nicole Zviedrite.

**Visualization:** Nicole Zviedrite.

**Writing – original draft:** Ferdous A. Jahan.

**Writing – review & editing:** Ferdous A. Jahan, Nicole Zviedrite, Hongjiang Gao, Faruque Ahmed, Amra Uzicanin.

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
