## [Decision Letter · Decision Letter 0]

7 Mar 2022

PONE-D-21-34949Causes, characteristics, and patterns of prolonged unplanned school closures prior to the COVID-19 pandemic – United States, 2011 – 2019PLOS ONE

Dear Dr. Zviedrite,

Thank you for submitting your manuscript to PLOS ONE. After careful consideration, we feel that it has merit but does not fully meet PLOS ONE’s publication criteria as it currently stands. Therefore, we invite you to submit a revised version of the manuscript that addresses the points raised during the review process.

We look forward to receiving your revised manuscript.

Kind regards,

Gerardo Chowell, PhD

Academic Editor

PLOS ONE

Journal Requirements:

Reviewers' comments:

Reviewer's Responses to Questions

**Comments to the Author**

1. Is the manuscript technically sound, and do the data support the conclusions?

Reviewer #1: Yes

2. Has the statistical analysis been performed appropriately and rigorously? 

Reviewer #1: Yes

3. Have the authors made all data underlying the findings in their manuscript fully available?

Reviewer #1: No

4. Is the manuscript presented in an intelligible fashion and written in standard English?

Reviewer #1: Yes

5. Review Comments to the Author

Reviewer #1: This is an important piece of research. The authors analyzed the "prolonged unplanned school closures" data collected by the CDC’s Community Interventions for Infection Control Unit (CI-ICU) and their contractors via online systematic search. The estimated number of student-days lost (as shown in Table 2: "Top twenty reasons for prolonged unplanned school closures (PUSC) according to the number of student-days lost, United States, 2011-2019") are important information both for education professionals and for disaster preparedness and response professionals.

Major comments:

1) Please clarify the following inconsistency of definition: is it 5d+ or 6d+? I guess – but this is not clear in the Methods section – is that the authors analyzed a subset of the data that contains unplanned school closures of 5d+ long. Among these 5d+ long unplanned school closures, the authors then analyze the association between selected characteristics and school closures of 6d+ using school closures of 5 days as the reference category (as explained in Line 105-108 and in the footnote to Table 3 in Line 217-221). This is interesting. But perhaps you need to better justify why you don’t analyze the whole dataset, or use schools with 0 d (or 0 to 5 d) of unplanned school closures as your reference category (for example). In other words, is it more important to know the adjusted odds ratio between those with unplanned school closures of 6+ d versus 5 d, than to know the adjusted odds ratio between 6+ d and 0 d, or that between 6+ d and 0-5 d? It is better to have a scientific reason to justify your choice of analysis.

• Line 70-71: “A closure duration of ≥5 days was categorized as PUSC.”

• Line 98-99: “The number of students enrolled pertains to all schools regardless of whether a school closed or not (i.e., schools with closes of 0 days, 1 – 4 days, or ≥5 days).

• Line 105-108: “The association between selected characteristics and school closures of ≥6 days among schools with PUSCs was assessed using schools as the unit of analysis…. The dependent variable in the regression models was coded as 0 (closure of 5 days) and 1 (closure of ≥6 days).”

2) In your limitation paragraph, please include an additional limitation: Given the source of data for this dataset (online systematic searches), some unplanned school closures that could have been identified using social media data might have been missed by analyzing online systematic searches alone. As shown in two recent studies that analyzed social media data in addition to the OSS data provided by the CDC’s CI-ICU (i.e., the authors of this manuscript), additional unplanned school closures that were not identified using OSS alone, were then identified on social media posts posted by the schools or school districts. This reviewer is confident that the authors are aware of these papers as they have cross-cleared those papers in the CDC. Perhaps the authors should cite them as well:

• Jackson AM, Mullican LA, Tse ZTH, Yin J, Zhou X, Kumar D, Fung ICH (2020). Unplanned closure of public schools in Michigan, 2015-2016: Cross-sectional study on rurality and digital data harvesting. Journal of School Health. 90(7):511-519. DOI: https://doi.org/10.1111/josh.12901

• Ahweyevu JO, Chukwudebe NP, Buchanan BM, Yin J, Adhikari BB, Zhou X, Tse ZTH, Chowell G, Meltzer MI, Fung ICH (2020). Using Twitter to Track Unplanned School Closures: Georgia Public Schools, 2015-17. Disaster Medicine and Public Health Preparedness. DOI: https://doi.org/10.1017/dmp.2020.65

Minor comments:

Line 60: Please write the full name of the virus when you introduce the acronym. Suggestion: please write: “severe acute respiratory syndrome coronavirus 2 (SARS-CoV-2)”

Liner 168: “(73,072,759or 81%)”: There should be space between the number and the word “or”.

Line 326: “categorized as prolonged, (lasting ≥5 days),”: You may delete the comma before “(lasting ≥5 days)”.

6. PLOS authors have the option to publish the peer review history of their article (what does this mean?). If published, this will include your full peer review and any attached files.

Reviewer #1: **Yes: **Isaac Chun-Hai Fung

---

## [Author Response · Author response to Decision Letter 0]

9 May 2022

Thank you for your thoughtful review. We have addressed your comments, and responded in a point-by-point fashion in an attached file. If agreed, we would like to include the added text in the Acknowledgements section to recognize your help in strengthening this work for publication. Thank you.

---

## [Decision Letter · Decision Letter 1]

13 Jul 2022

Causes, characteristics, and patterns of prolonged unplanned school closures prior to the COVID-19 pandemic – United States, 2011 – 2019

PONE-D-21-34949R1

Dear Dr. Zviedrite,

We’re pleased to inform you that your manuscript has been judged scientifically suitable for publication and will be formally accepted for publication once it meets all outstanding technical requirements.

Kind regards,

George Vousden

Staff Editor

PLOS ONE

Additional Editor Comments (optional):

PLOS ONE does not permit thanking of reviewers in the Acknowledgements section; please remove this section from your Acknowledgements. For further detail, please see here: https://journals.plos.org/plosone/s/submission-guidelines#loc-acknowledgments

The reviewer has provide further suggestions to the sentence structure of the Acknowledgements; compliance with this recommendation is optional.

Reviewers' comments:

Reviewer's Responses to Questions

**Comments to the Author**

1. If the authors have adequately addressed your comments raised in a previous round of review and you feel that this manuscript is now acceptable for publication, you may indicate that here to bypass the “Comments to the Author” section, enter your conflict of interest statement in the “Confidential to Editor” section, and submit your "Accept" recommendation.

Reviewer #1: All comments have been addressed

2. Is the manuscript technically sound, and do the data support the conclusions?

Reviewer #1: (No Response)

3. Has the statistical analysis been performed appropriately and rigorously? 

Reviewer #1: (No Response)

4. Have the authors made all data underlying the findings in their manuscript fully available?

Reviewer #1: (No Response)

5. Is the manuscript presented in an intelligible fashion and written in standard English?

Reviewer #1: (No Response)

6. Review Comments to the Author

Reviewer #1: The authors have adequately addressed my comments. It is ok to include my name in the Acknowledgements. Thank you.

Minor comments:

Line 424-425 (track change version): “For her contributions to data visualization, we thank Sarah Moreland”. The sentence structure is strange. The part of the sentence before the comma describes the subject of the sentence. In this case, the subject is “we”. But in fact, what the authors would like to say is that: “We thank Sarah Moreland for her contributions to data visualization”. The authors should probably stick to their original sentence. The revised sentence in the R1 version is grammatically incorrect.

7. PLOS authors have the option to publish the peer review history of their article (what does this mean?). If published, this will include your full peer review and any attached files.

Reviewer #1: **Yes: **Isaac Chun-Hai Fung

---

## [Editor Report · Acceptance letter]

21 Jul 2022

PONE-D-21-34949R1 

Causes, characteristics, and patterns of prolonged unplanned school closures prior to the COVID-19 pandemic – United States, 2011 – 2019 

Dear Dr. Zviedrite:

I'm pleased to inform you that your manuscript has been deemed suitable for publication in PLOS ONE. Congratulations! Your manuscript is now with our production department. 

Kind regards, 

on behalf of

Dr. George Vousden 

Staff Editor

PLOS ONE